# Premorbid Psychological Factors Associated with Long-Term Postoperative Headache after Microsurgery in Vestibular Schwannoma—A Retrospective Pilot Study

**DOI:** 10.3390/brainsci13081171

**Published:** 2023-08-07

**Authors:** Mareike Thomas, Stefan Rampp, Maximilian Scheer, Christian Strauss, Julian Prell, Robby Schönfeld, Bernd Leplow

**Affiliations:** 1Department of Medical Psychology, Center for Psychosocial Medicine, University Medical Center Hamburg-Eppendorf, Martinistrasse 52, 20246 Hamburg, Germany; m.thomas@uke.de; 2Department of Psychology, Martin-Luther-Universität Halle-Wittenberg, Emil-Abderhalden-Straße 26–27, 06108 Halle, Germany; 3Department of Neurosurgery, University Hospital Halle, Ernst-Grube-Straße 40, 06120 Halle, Germany; 4Department of Neurosurgery, University Hospital Erlangen, Schwabachanlage 6, 91054 Erlangen, Germany; 5Department of Neuroradiology, University Hospital Erlangen, Schwabachanlage 6, 91054 Erlangen, Germany

**Keywords:** vestibular schwannoma, postoperative headache, premorbid psychological factors, microsurgery

## Abstract

Associations between premorbid psychological factors and postoperative headache (POH) after microsurgical treatment via the retrosigmoid approach for vestibular schwannoma (VS) were investigated in this retrospective single-center study. A total of 101 VS patients completed the Rostock headache questionnaire (RoKoKo), the hospital and anxiety scale (HADS-D), and the screening for somatoform disorders (SOMS-2), all of which were used as short self-assessed questionnaires. Fifty-four patients with POH were compared with 47 non-POH patients in terms of premorbid psychological factors, somatization tendencies, and psychological burden using the chi^2^-test and Mann–Whitney *U*-test. Regression analyses were conducted to assess the weighted contribution of psychological and procedural factors to POH. In individuals with POH, mental ailments, preexisting headaches, premorbid chronic pain syndromes, and higher somatization tendencies were found to be significantly more common. POH was predicted by the number of premorbid psychosomatic symptoms, preexisting mental ailments, and premorbid chronic pain syndromes. Depression and anxiety were predicted by low emotional stability. Additionally, the number of premorbid psychosomatic symptoms predicted depression, anxiety, and overall psychological burden. It was observed that the reported symptoms of headache might fit into the classification of chronic postsurgical pain (CPSP) rather than being classified as secondary headaches after craniotomy. Premorbid psychological factors were found to play an important role in the emergence of POH in VS, particularly after microsurgery via the retrosigmoid approach. Therefore, it is suggested that psychological screening be incorporated into the treatment process.

## 1. Introduction

Vestibular schwannoma (VS) is a benign tumor originating from Schwann cells. Located intra- and extrameatal, hearing loss, dizziness, and tinnitus are common symptoms resulting from the tumor’s affection of the vestibulocochlear nerve. A current prospective study suggests that those complications are unlikely to be significantly modified by treatment modality, i.e., observation, microsurgery, or radiosurgery [1], but previous investigations showed an increased rate, especially after microsurgical treatment [2,3]. A common complication of the retrosigmoid surgical approach is the emergence of postoperative headache (POH) [4]. Acute POH should not last longer than three months, whereas persistent POH attributed to craniotomy is characterized by more frequent headaches of more than three months duration [5]. The International Headache Committee classified POH as a secondary headache that occurs within seven days post-surgery. About 25 percent of patients with acute POH also develop persistent POH [5]. In fact, most studies on post-suboccipital craniotomy headaches address VS [6].

Not surprisingly, the vast majority of past research on POH in the neurosurgical field has been focused on discussing medical reasons as well as procedural factors, such as intracranial bone dust [7,8,9,10,11], fibrous adhesions [7,8,9,10,11,12,13,14,15], occipital nerve injury [7,10,13,16,17,18,19], and scar tissue [8,10,11,13,16], neurological inflammation [7], dural/meningeal irritation [16,17], cerebrospinal fluid leaks [10], or vascular causes [7,9,16,18]. Studies with a focus on different surgical methods have found a lower incidence of POH for craniotomies in comparison with craniectomy [11,20,21], but the findings in terms of cranioplasty have been contradictory [12,15,22].

Psychological symptoms in VS patients, like depression and anxiety, have all been examined for the postoperative state [9,23,24]. Elevated levels of anxiety and depression have been found in a study by Brooker and colleagues [23], with no differences with respect to management group (microsurgery vs. radiation vs. wait and scan). Rimaaja and colleagues [9] found an association between POH and higher scores on the Beck’s Depression Scale (BDI-II), and in a study by Farace et al. VS-patients showed an increased suicide rate [24]. In contrast to these studies, Kalayam et al., [25] reported on 803 consecutive inpatient psychiatric admissions in which psychiatric symptoms began preoperatively and remained for long periods after surgery. They implied that these transient symptoms, including mood disorders, might originate from the disruption of brainstem structures due to the tumor. In accordance with these findings, Li et al. [26] reported higher preoperative levels of anxiety and depression. This psychological burden was associated with a range of risk factors: time since diagnosis, number of symptoms, headache, vertigo, nausea, and/or vomiting. Secondary chronic headaches are also associated with high psychological distress and high levels of neuroticism compared to the general population [27].

The aim of this study is to: (1) characterize premorbid psychological factors in patients with VS who develop POH; (2) determine the weighted contribution of these variables to the occurrence of POH; and (3) find premorbid predictors for postoperative anxiety and depression in individuals with POH. Results might be helpful for the management of POH in VS by shining light on psychological targets that can be addressed before surgery to prevent the emergence of severe POH.

## 2. Materials and Methods

### 2.1. Study Design and Population

This study followed a retrospective approach. Inclusion criteria were diagnosis, a minimum age of 18 at diagnosis, surgery via retrosigmoid approach, and German as a native language. Patients who underwent previous surgery or radiation and/or suffered from recurrent VS, as well as patients with oncological diagnoses and neurofibromatosis type 2, were excluded. The study was started in early 2020, which resulted in the recruitment of patients being influenced by the COVID-19 pandemic. Participants were approached directly at the chief resident consultation at University Hospital Halle and after the start of the national lockdown via a call for participation by the *Vereinigung Akustikus Neurinom e.V.* (a non-profit patient self-help organization) to complete an online survey (*SoSciSurvey*). Both surveys ran identically. All participants provided written consent prior to study participation. The study was approved by the Ethics Committee of the University Hospital Halle (No. 2020-008).

### 2.2. Measures

POH was defined as a long-term, frequent headache that lasted for more than 3 months after microsurgery. We assessed POH using the Rostock headache questionnaire (RoKoKo) to assign the symptoms to migraine (constant), tension-type headache, or other headaches [28]. It covers different aspects of headaches, such as frequency, pain characteristics, and location. Items to measure demographic details (e.g., age and gender) and psychological factors, such as the premorbid existence of psychological diagnoses (e.g., major depression disorder) or mental ailments (e.g., sleep disturbances), as well as the existence of premorbid chronic pain syndromes (e.g., chronic back pain), and information with respect to treatment (surgical approach, craniotomy vs. craniectomy), were assessed via a short self-created questionnaire. These items were dichotomous (yes/no) and could be specified (e.g., “Which psychological disorder was diagnosed?”). The tumor size was reported as Koos grade [29]. Premorbid somatization tendencies were measured by the screening for somatoform disorders (SOMS-2) test, which has been widely used and validated in patient populations with psychosomatic disorders [30]. In the first section, participants are asked to report physical symptoms they have suffered, either temporarily or continuously, in the previous two years. In this study, patients were asked to report on those symptoms for a period of two years prior to the VS diagnosis. These symptoms should have been significantly disturbing to their well-being or their personal lifestyle, and doctors should not have found a clear cause for them. A list of 53 somatoform symptoms, five only for women and one for men, is described (diagnostic and statistical manual of mental disorders, Version DSM-IV-TR criteria and ICD-10 criteria). The second section comprises 15 questions aimed at evaluating disability, the number of consultations due to symptoms, and the inclusion/exclusion criteria for all somatoform disorders. To assess current anxiety and depression symptoms, the researchers employed the hospital anxiety and depression scale (HADS-D) [31,32]. The HADS-D consists of two separate scales, one for anxiety and the other for depression, each yielding scores between 0 and 21. Participants were asked to report symptoms experienced during the past week, and higher scores indicate higher levels of anxiety and depression. While this scale alone cannot serve as a diagnostic tool, scores exceeding 7 suggest the presence of an affective disorder, warranting further evaluation. For evaluating emotional stability as a personality trait, the corresponding subscale of the ten item personality inventory (TIPI-G) was used. This concise questionnaire provides an efficient approximation of the five-factor model of personality, making it particularly useful when time for evaluation is limited [33].

### 2.3. Statistical Analysis

Statistical analysis was carried out using SPSS for Windows Version 25 (IBM, Armonk, NY, USA). Demographic characteristics, including sex at birth and age, premorbid chronic pain syndromes and mental ailments, and scores on the SOMS-2, HADS-D, and TIPI, were compared between participants with and without POH. Distribution was assessed by the Kolmogorov–Smirnov test. For clarity, all summary statistics are reported as means with a standard deviation (SD). Categorical features were summarized with frequency counts and percentages. To identify group differences between participants with and without POH, dichotomous data were compared using chi-squared tests. Metric data were compared using Mann–Whitney U tests. For comparisons between different POH groups, Kruskall–Wallis tests were used. Statistical significance was accepted at a *p*-value of 0.05. In the online survey, the option “I don’t know” was given for questions concerning the surgical approach (e.g., craniectomy or craniotomy, positioning during surgery). These answer options were counted as “missing” and were not included in the statistical analysis for group differences.

To understand the weighted contribution of psychological variables to POH, a binary logistic regression was conducted. A total of three models were established: Model 1 was formed to account for the potential effects of variables that have been significant predictors in previous studies, such as age at onset, sex at birth, time since surgery, and tumor size. The dichotomous psychological variables premorbid psychological ailments, premorbid chronic pain syndromes, and emotional stability as a psychological trait were included in Model 2. Then, a final stepwise binary logistic regression using the Likelihood ratio Chi^2^-criterion was performed using all variables mentioned above. Current depression and anxiety scores were not taken into account, as only premorbid and preprocedural factors are of interest. This statistical approach was based on the work of Peña et al. [34]. The goodness-of-fit was assessed by the Hosmer–Lemeshow test. To predict current psychological burden in the form of HADS-D scores, stepwise multiple linear regressions were conducted, including demographic and psychological factors as independent variables. A probability of *F* = 0.05 was used as the entry criterion. Goodness-of-fit was assessed by an adjusted *R*^2^ as suggested by Cohen [35].

## 3. Results

### 3.1. Demographic, Disease and Surgery Related Variables

A total of 101 patients were included (27 at University Hospital Halle, 74 via online survey), with 54 participants (53.5%) reporting the existence of POH. A summary of group comparisons can be found in Table 1. Age at onset did not significantly differ between POH and non-POH groups (*p* = 0.074), whereas participants with POH were significantly younger at the time of the survey (*p* = 0.039). In terms of sex, there were significantly fewer male patients in the POH group (POH, 27.8% male; non-POH, 48.9% male; *p* = 0.029). There was no significant difference in time since surgery (*p* = 0.096). Groups did not differ concerning facial nerve paresis (*p* = 0.267) and hearing loss after surgery (*p* = 0.335). With respect to tumor size, a total of seven participants from the online cohort were excluded from the analysis. A group comparison of tumor size among the remaining participants showed no significant difference in Koos grade (*p* = 0.222). For analysis concerning craniectomy and positioning during surgery, 14 and 10 patients were excluded, respectively. POH patients did not differ from non-POH patients concerning craniectomy (*p* = 0.095) or positioning during surgery (*p* = 0.654).

### 3.2. POH Characteristics

To identify the characteristics of POH, we used the *RoKoKo,* which differentiates four different types of headaches: paroxysmal (*n* = 13), constant (*n* = 15), paroxysmal and constant (*n* = 19), and other than described (*n* = 8). A summary of headache types and pain location is shown in Figure 1. Patients were to characterize their pain; multiple answers were possible. Five patients experiencing paroxysmal POH described the pain as mostly unilateral, two as mostly bilateral, three reported alternating locations during an attack, and one patient reported different locations between attacks. Most patients in this group reported occipital (*n* = 4), central (*n* = 4), or frontal (*n* = 3) pain. In the constant pain group, the majority of patients reported mostly unilateral pain (*n* = 8) in the temporal (*n* = 5) or central (*n* = 4) regions. Patients experiencing paroxysmal and constant POH reported mostly unilateral pain (*n* = 16), in some cases alternately during (*n* = 3) or between attacks (*n* = 2). These patients described the pain location mainly as temporal (*n* = 10) or occipital (*n* = 8). In the “other than described” group, 8 participants reported mostly unilateral pain, and 5 described POH as mostly bilateral.

Figure 2 summarizes absolute frequencies for the type of POH and pain character. Participants with paroxysmal POH described lancinating (*n* = 7), pulsing (*n* = 4), gnawing (*n* = 3), and burning (*n* = 1) sensations. In the constant pain group, mostly pulsing (*n* = 8) and dull (*n* = 9) pain was reported, followed by a lancinating (*n* = 6) and burning (*n* = 1) character. Lancinating (*n* = 19) and dull (*n* = 14) pain occurred in most patients with both paroxysmal and constant pain. The “other than described” pain group reported mostly lancinating (*n* = 8) and dull pain (*n* = 6) as well.

### 3.3. Premorbid Pain Syndromes and Premorbid Psychological Variables

Table 2 provides a comprehensive summary of group comparisons between patients with POH and those without POH as well as descriptive data on premorbid pain syndromes, psychological ailments, and disorders. Premorbid chronic pain syndromes were reported in 44.4% of POH patients and 12.8% of participants in the non-POH group (*p* < 0.001). The most frequently mentioned syndromes were premorbid headache syndromes and chronic back pain, followed by stomach and bowel problems and face and knee pain (Table 2). In the POH group, participants reported significantly more premorbid mental ailments (*p* = 0.001), such as difficulty falling asleep or sleeping problems through the night, undefined physical exhaustion, panic attacks, rumination, and in one case even a suicide attempt. No group differences in terms of the existence of premorbid psychological diagnoses were found (*p* = 0.106). The mentioned diagnoses included Major depressive episodes, burnout syndrome, acute stress disorder, anxiety disorder, and posttraumatic stress disorder (Table 2). Premorbid headache was significantly more frequent in the POH group (*p* = 0.048).

Table 3 and Figure 3 show mean scores of psychological questionnaires for groups. SOMS-2 scores to measure premorbid somatization tendencies showed significant group differences for symptom count score (*p* < 0.001), somatization index ICD-10 (*p* = 0.016), and somatization index DSM-IV (*p* = 0.001) scores, with POH patients reaching higher scores than non-POH patients. The same is shown for current mood in HADS-D scores, with POH patients reporting significantly higher scores in the depression (*p* = 0.034) and anxiety subscores (*p* = 0.014) and the total score (*p* = 0.008). No group differences were found for emotional stability (*p* = 0.383).

Different headache groups according to the RoKoKo did not differ with respect to symptom count score (*p* = 0.498), somatization index ICD-10 (*p* = 0.545), somatization index DSM-IV (*p* = 0.148), depression (*p* = 0.222), anxiety (*p* = 0.537), HADS-D total score (*p* = 0.535), and emotional stability (*p* = 0.314), Figure 4.

### 3.4. Regression Analysis

The results of all regression models conducted for POH are shown in Table 4. Missing cases for the variables tumor size (Koos) and positioning during the procedure were excluded. Thus, 80 cases were considered for all three models. Model 1, including demographics and procedural characteristics, was not statistically significant, *χ*^2^(6) = 9.78, *p* > 0.05 and consequently does not bear significant predictive value for POH. The psychological measures Model 2, however, was statistically significant (*χ*^2^(4) = 22.83, *p* < 0.001 (*R*^2^ = 0.25)), with preexisting chronic pain syndromes being a significant predictor. Goodness-of-fit was assessed using the Hosmer–Lemeshow test, indicating a good model fit *χ*^2^(8) = 9.53, *p* > 0.05. In the stepwise regression, which included all variables from Models 1 and 2, variables were added to the model based on their contribution. In the stepwise regression, which included all variables from Models 1 and 2, variables were added to the model based on their contribution. This final model (*χ*^2^(3) = 24.23, *p* < 0.001, (*R*^2^ = 0.27)) retained age at onset, premorbid mental ailments, and premorbid chronic pain syndromes as significant predictors for POH (Hosmer–Lemeshow test *χ*^2^(7) = 10.64, *p* > 0.05). According to these results, a younger age at onset as well as the existence of premorbid mental ailments and chronic pain syndromes were shown to be predictive factors for the emergence of POH.

Regression analysis for HADS-D scores can be found in Table 5. A low level of emotional stability was able to predict the HADS-D total score with statistical significance, *F*(1, 47) = 36.21, *p* < 0.001. The *R*^2^ for the overall model was 0.44 (adjusted *R*^2^ = 0.42), indicative of a high goodness-of-fit, according to Cohen [35]. A significant predictor for the HADS-D depression score was low emotional stability and male sex (*F*(1, 47) = 12.13, *p* = 0.001, *R*^2^ = 0.27) with a moderate goodness-of-fit (adjusted *R*^2^ = 0.24). HADS-D anxiety scores could be significantly predicted by a high number of symptoms and low emotional stability (*F*(1, 47) = 48.13, *p* < 0.001, *R*^2^ = 0.55), showing a high goodness-of-fit (adjusted *R*^2^ = 0.53).

## 4. Discussion

In this pilot study, we analyzed the relationship between long-term POH and premorbid psychological variables as well as the implications of premorbid psychological factors on depression and anxiety levels in patients with VS. The major finding of our investigation was the association between premorbid psychological factors and POH in patients with VS after microsurgical treatment via the retrosigmoid approach. The results indicate that patients with POH are more likely to have premorbid mental ailments like sleep disturbances, panic attacks, and rumination, as well as premorbid chronic pain syndromes such as chronic back pain or primary headache syndromes. These psychological factors seem to weigh more than procedural factors like positioning during surgery. The number of premorbid psychosomatic symptoms is a significant predictor of psychological burden in POH patients, whereas depression and anxiety are also influenced by low levels of emotional stability.

Our results are in accordance with preexisting literature regarding premorbid psychological symptoms in VS patients [25,26]. Those results could be extended by findings with regards to the existence of POH. While previous studies focused on psychological burden in the general population of VS patients, our study showed that patients suffering from POH exhibited more premorbid mental ailments than non-POH patients. The regression analysis indicates that a high number of premorbid psychosomatic symptoms, such as diarrhea and sweaty hands, predict overall psychological burden in POH patients. Further, disease-based structural issues could also contribute to psychologic symptoms that present preoperatively [25]. Li et al. [26] also found a high number of symptoms associated with preoperative psychological burden in patients with VS. It may be possible that this burden continues beyond the period after surgery. Future studies should investigate whether psychological symptoms in VS could be normalized at least to a certain degree and therefore reduce stigma around those symptoms while legitimizing a complementary psychological approach to disease management.

An association between current HADS scores and headache impairment was demonstrated by Carlson et al. [36]. Similarly, our findings indicate that individuals with POH are more likely to experience higher scores related to depression and anxiety. However, when predicting headaches, we focused solely on premorbid factors, considering that headaches and depression often co-occur. It has been observed that individuals with migraine have a higher likelihood of having a psychiatric comorbidity compared to the general population [37]. Additionally, high scores in neuroticism, which indicate low emotional stability, are also linked to depression. This association between neuroticism and depression has been observed not only in the general population [38] but also among migraine patients [39]. Within the context of the biopsychosocial model [40], it remains unclear which condition influences the other, and it is possible that there is a mutual interaction between them.

In our study, we classified POH as a secondary headache syndrome. In the new international classification of diseases (ICD-11), another category is being introduced: chronic postsurgical pain (CPSP), which could also fit POH. CPSP is usually described as neuropathic pain with a traumatic cause, and risk factors include female sex at birth, young age, the existence of preoperative pain conditions, and perioperative factors such as duration and type of surgery [41]. Depression, psychological vulnerability, and stress have been shown to be associated with CPSP [42]. Neuropathic pain is often described as lancinating or shooting in character [43], which was the case for 74% of POH patients in this study. It also occurs more prominently in paroxysms, which applies to 15/54 of our POH patients and 19/54 POH patients with paroxysmal and constant headache. Interpreting POH as shown in this study, the current findings would fit the previously listed research in CPSP. With respect to perioperative factors, the majority of POH patients report mostly unilateral occipital and temporal pain. This could indicate lesions of the occipital nerve as a biological factor in accordance with previous findings [7,10,13,16,17,18,19]. As the patients in the cohort have been operated on with the retrosigmoid approach, occipital neuralgia may occur as a result of nerve entrapment. 

A comparison between patients with and without POH revealed a significantly higher incidence of headaches before surgery (38.9% vs. 21.3%, *p* = 0.048). It is therefore conceivable that, at least in some of these patients, POH may be a continuation of preoperative symptoms. However, this also implies that surgical treatment was not able to abolish these, resulting in postoperative chronic headaches largely indistinguishable from patients with newly developed POH. This latter group amounted to 61.1% of POH patients, underscoring that POH is a relevant clinical issue partly independent of preoperative status. Further studies should investigate details of qualitative characteristics as well as the evolution over time of preexisting headaches to potentially identify a corresponding subgroup. It is also worth noting that some participants mentioned experiencing premorbid headache syndromes when asked about premorbid pain syndromes, which could potentially impact the results regarding this variable. However, since group differences for all SOMS-2 measures were still evident between the POH and Non-POH groups, we can infer that these differences might be unrelated to the presence of premorbid headache as reported in the specified variable.

The biopsychosocial model [40] describes pain as a multidimensional condition with interactions between biological, psychological, and social factors. Though it seems clear that depression and anxiety can be the result of pain conditions, the influence is bidirectional: psychological disorders can also predispose to chronic pain. The model suggests that psychological factors may influence the development of chronic pain syndromes but also hold an important role in treatment as resilience factors [40]. Sleep disturbances could therefore be seen as biological factors. In this study, we classified sleeping difficulties as a psychological variable, as sleep disorders are commonly comorbid with and often treated as behavioral disorders [41].

### Limitations

The results of this study must be interpreted against the background of some methodological weaknesses. First, conducting a study via online surveys has disadvantages. The lack of an interviewer leaves no space for clarification of unfamiliar or ambiguous terms. Also, the responses of those who do not have access to the internet, especially elderly patients, will not be captured (respondent bias). Anonymous participation is also prone to fraud [42].

Employing a retrospective approach in this study introduces the possibility of memory bias. The average duration between the survey and surgery was relatively extensive. This extended time frame offers the advantage of examining the long-term effects of POH. However, it also creates an environment conducive to distorted memories of the period preceding the surgery. Anxiety and depression can influence memory biases, leading to misrepresentations of daily emotional experiences and a tendency to recall more negative affect [43]. Though the study by Mathersul and Ruscio [43] referred to clinical populations of major depression and general anxiety disorder, the POH group shows an average score of 8 in the HADS-D questionnaire, which is the suggested cut-off for further clinical examination of symptoms. Annunziata et al. even suggest a cut-off of 7 on the anxiety scale for cancer patients [44]. This may be a sign of anxiety above the subclinical level, indicating a prolonged negative emotional state in at least some of the participants. Thus, results for the SOMS-2 questionnaire might also be inaccurate for some of the patients.

Overall, the distinction between psychological and biological variables has not been sharp enough. Sleep disturbances and physical exhaustion can be psychological symptoms but can also be related to somatic illnesses or prodromal symptoms. Because of the retrospective approach, there were no objective measures for these variables. We simply had to trust the reliability of the respondents.

## 5. Conclusions

POH is a major problem for patients after microsurgical treatment in VS. Our study suggests that younger patients with female sex at birth, premorbid somatization tendencies, and mental ailments might be at a higher risk of developing POH. The results indicate the classification of POH as CPSP rather than a secondary headache syndrome, which can be preceded by high somatization tendencies and mental ailments. A high number of premorbid psychosomatic symptoms predicts postoperative psychological burden in patients with POH, and depression is also associated with neuroticism. Future studies should verify our findings to be generalizable with a prospective approach to avoid memory bias for premorbid factors. There should also be a sharper distinction between psychological and biological factors. Further, larger cohorts with respect to different surgical procedures should be studied. As a result, the preoperative examination should include a standardized psychological examination. Finally, VS patients at risk for POH should get the opportunity to choose psychological support during treatment and the follow-up period.

## Figures and Tables

**Figure 1 brainsci-13-01171-f001:**
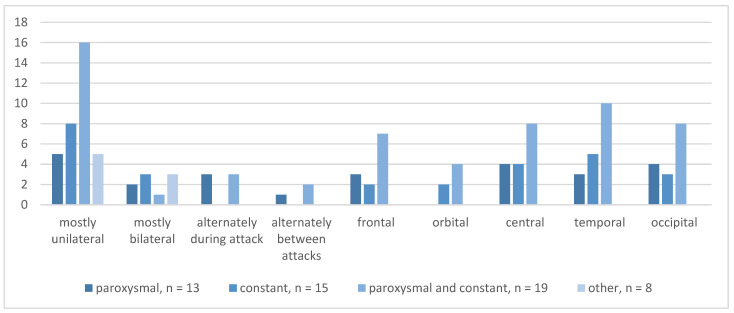
Absolute frequencies for type of POH and pain location.

**Figure 2 brainsci-13-01171-f002:**
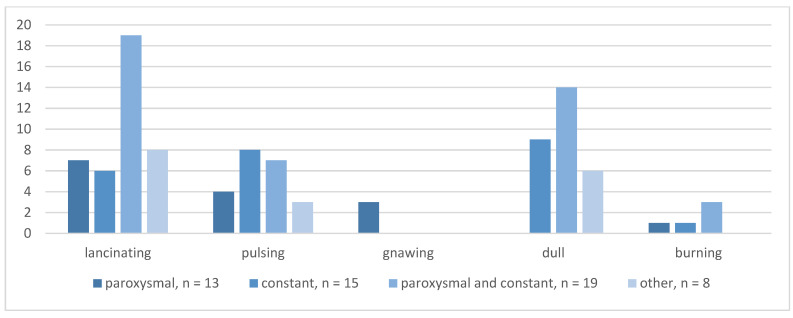
Absolute frequencies for type of POH and pain character.

**Figure 3 brainsci-13-01171-f003:**
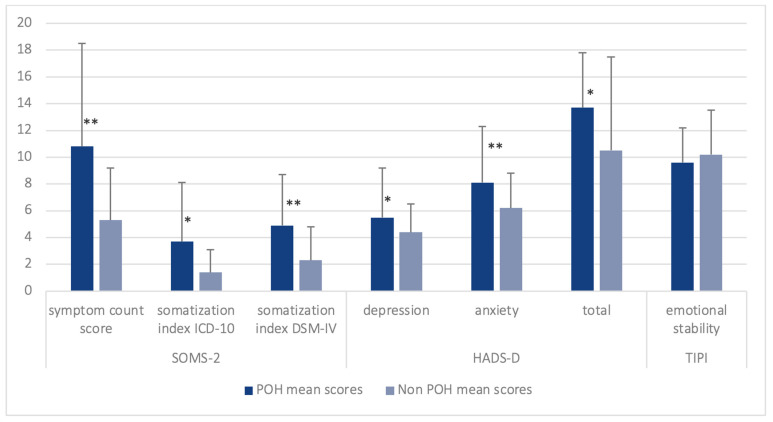
Mean scores and standard deviation of SOMS-2, HADS-D and TIPI subscales for groups * *p* < 0.05, ** *p* < 0.001.

**Figure 4 brainsci-13-01171-f004:**
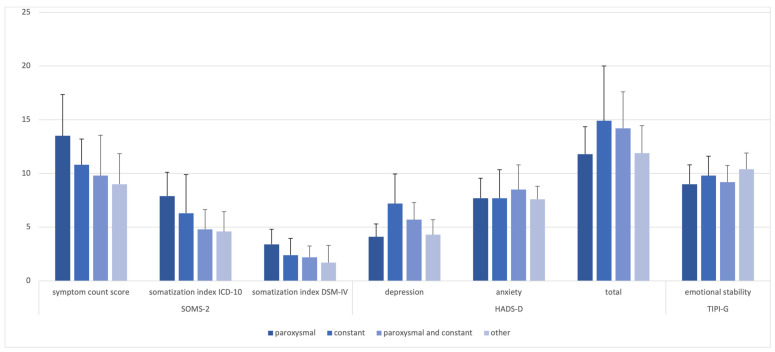
Mean scores for psychological questionnaires between headache groups.

**Table 1 brainsci-13-01171-t001:** Statistical analysis of mean differences between POH- and Non POH group for demographic, disease and surgery related variables.

	Central Tendency		*p*-Score POH vs. Non POH
	POH, *n* = 54	Non POH, *n* = 47	
male sex at birth, no. (%)	15 (27.8)	23 (48.9)	0.029
age at survey, mean (SD)	54.5 (12.2)	58.5 (11.8)	0.039
age at onset, mean (SD)	46.3 (10.2)	49.0 (11.7)	0.074
time since surgery in months, mean (SD)	121.6 (109.3)	139.7 (95.6)	0.096
Koos, no. (%)			0.222
1	8 (14.8)	5 (10.6)	
2	14 (25.9)	10 (21.3)	
3	20 (37.0)	14 (29.8)	
4	9 (16.7)	13 (27.7)	
missing	2 (3.7)	5 (10.6)	
facial nerve paresis, no. (%)	12 (22.2)	15 (31.9)	0.267
missing	1 (1.9)	1 (2.1)	
hearing loss after surgery, no. (%)	27 (50.0)	28 (59.6)	0.335
craniectomy, no. (%)	23 (42.6)	30 (63.8)	0.095
missing	10 (18.5)	4 (8.5)	
positioning during surgery, no. (%)			0.654
semi-sitting	37 (68.5)	34 (72.3)	
supine	8 (17.0)	8 (17.0)	
missing	5 (10.6)	5 (10.6)	

**Table 2 brainsci-13-01171-t002:** Statistical analyses of mean differences between POH and Non-POH groups and complementary descriptive data of specified premorbid characteristics for groups.

Variable *	Group		*p*-Score POH vs. Non POH
premorbid pain syndromes	POH, *n* = 24 (44.4%)	Non POH, *n* = 6 (12.8%)	<0.001
chronic back pain	11	2	-
headache	17	2	-
face pain	2	0	-
knee pain	1	0	-
stomach and bowel problems	4	1	-
premorbid mental ailments	POH, *n* = 25 (46.3%)	Non POH, *n* = 8 (17.0%)	<0.001
difficulty falling asleep/ problems sleeping through the night	20	4	-
physical exhaustion	3	2	-
panic attacks	3	1	-
rumination	3	0	-
psychosomatic stomach pain	0	1	-
suicide attempt	1	0	-
premorbid psychological diagnosis	POH, *n* = 15 (27.8%)	Non POH, *n* = 7 (14.9%)	0.106
Major Depression	11	2	-
Burnout	2	2	-
acute stress disorder	3	0	-
anxiety disorder	1	1	-
posttraumatic stress disorder	0	2	-
premorbid headache syndromes	POH, *n* = 21 (38.9%)	Non POH, *n* = 10 (21.3%)	0.048

* multiple answers were possible, note. - = not calculated.

**Table 3 brainsci-13-01171-t003:** Statistical analyses of mean differences for psychological questionnaires between POH and Non-POH groups.

	Central Tendency		*p* Value POH vs. Non POH
	POH, *n* = 54	Non POH, *n* = 47	
SOMS-2			
symptom count score, mean (SD)	10.9 (8.0)	5.6 (3.8)	<0.001
somatization Index ICD-10, mean (SD)	2.5 (2.5)	1.3 (1.3)	0.016
somatization Index DSM-IV, mean (SD)	5.9 (4.8)	2.9 (2.5)	0.001
HADS-D			
depression, mean (SD)	5.6 (3.9)	3.9 (2.2)	0.034
anxiety, mean (SD)	7.9 (4.3)	5.8 (2.7)	0.014
total score, mean (SD)	13.5 (7.3)	9.8 (4.2)	0.008
TIPI-G			
emotional stability, mean (SD)	9.6 (3.3)	10.3 (2.5)	0.383

Abbreviations: SOMS-2, Screening for Somatoform Disorders; HADS-D, Hospital Anxiety and Depression Scale—German version; TIPI-G, Ten Item Personality Inventory.

**Table 4 brainsci-13-01171-t004:** Regression analysis for variables associated with the existence of POH.

	Model 1: Demographics and Procedural Characteristics	Model 2: Psychological Measures	Model 3: Stepwise Regression
Intercept	0.93 (1.90)	−1.89 (1.46)	−1.58 (1.20)
sex at birth	0.95 (0.53)	-	-
age at onset	−0.04 (0.03)		−0.51 (0.03) *
tumor size (Koos)	−0.20 (0.27)	-	-
positioning during surgery	0.03 (0.73)	-	-
time since treatment (months)	−0.003 (0.003)	-	-
preexisting headache	0.45 (0.40)		-
premorbid mental ailments	-	1.16 (0.68)	1.85 (0.69) *
premorbid chronic pain syndromes	-	1.45 (0.67) *	1.55 (0.68) *
symptom count score ^†^	-	0.48 (0.41)	-
emotional stability	-	0.41 (0.10)	-

* *p* < 0.05. ^†^ logarithmised. Values shown as coefficient (standard error).

**Table 5 brainsci-13-01171-t005:** Stepwise multiple linear regression models to predict current psychological burden (HADS-D scores) for individuals with POH.

	HADS-D Depression	HADS-D Anxiety	HADS-D Total
Intercept	14.80 (3.50)	13.01 (3.00)	27.89 (2.50)
sex at birth	−2.26 (1.12) *	-	-
age at onset	-	-	-
time since sugery (months)	-	-	-
premorbid mental ailments	-	-	-
premorbid chronic pain syndromes	-	-	-
symptom count score ^†^	-	1.29 (0.60) *	-
emotional stability	−0.54 (0.15) **	−0.80 (0.15) **	−1.48 (0.25) **

* *p* < 0.05. ** *p* < 0.001. ^†^ logarithmised. Values shown as coefficient (standard error).

## Data Availability

The data presented in this study are available on request from the corresponding author. The data are not publicly available due to privacy reasons.

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
