# Peer review of "Premorbid Psychological Factors Associated with Long-Term Postoperative Headache after Microsurgery in Vestibular Schwannoma—A Retrospective Pilot Study"

_brainsci, 2023, doi:10.3390/brainsci13081171_

Round 1
Reviewer 1 Report
The authors conducted a retrospective study to investigate associations between premorbid psychological factors and postoperative headache (POH) after microsurgical treatment via retrosigmoid approach for vestibular schwannoma (VS). This is an interesting study that highlighted premorbid psychological factors seem to play an important role in the emergence of POH in VS, especially after microsurgery via retrosigmoid approach. However, there are some points to be further improved as well.
1. Line 176, where is Table 5? It is not present in the manuscript.
2. Lines 195-196, does this sentence correspond to Figure 1? Where can we find the premorbid chronic pain syndromes proportions and statistical analysis between the POH group and nonPOH group? The data described must be presented specifically in the manuscript.
3. Lines 205-211, the statistical p-values described in this section for Table 3 must be included within Table 3.
4. The asterisks (*) in Figures 3 and 4 have varying degrees of darkness, and their positions are not centered. It is recommended to redraw the figures using dedicated graphing software.
Minor editing of English language required.
Author Response
Dear Reviewer,
Thank you very much for taking the time to review our manuscript. Your comments were very helpful, and I tried to incorporate everything that you mentioned.
I restructured the results section and added subheadings. I also divided Table 1 into two tables (Table 1. Statistical analysis of mean differences between POH- and Non POH group for demographic, disease and surgery related variables and
Table 2. Statistical analyses of mean differences between POH and Non POH groups and complementary descriptive data of specified premorbid characteristics for groups.). I added the missing figure and added another figure for differences between headache groups (Figure 3. Mean scores and standard deviation of SOMS-2, HADS-D and TIPI subscales for groups
Figure 4. Mean scores for psychological questionnaires between headache groups). I also included the p-scores in the Tables.
I also changed the asterisks to have the same colour.
All of the changes I made are highlighted.
We look forward to hearing from you in due time regarding our submission and to respond to any further questions and comments you may have.
Reviewer 2 Report
Well done and honest conclusions:A comparison between patients with and without POH revealed a significantly 318
higher incidence of headaches already before surgery (38.9% vs. 21.3%, P = 0.048. It is
therefore conceivable that at least in some of these patients, POH may be a continuation
of preoperative symptoms. However, this also implies that surgical treatment was not able
to abolish these, resulting in postoperative chronic headache largely indistinguishable
from patients with newly developed POH.
Author Response
Dear Reviewer,
We appreciate the time and effort that you and the reviewers have dedicated to providing your valuable feedback on our manuscript.
I restructured the results section due to comments from another reviewer and highlighted all the changes I made in the manuscript.
We look forward to hearing from you in due time regarding our submission and to respond to any further questions and comments you may have.
Reviewer 3 Report
Thank you for allowing me to review the work of the authors regarding this very interesting topic. Despite the major advances in NeuroSurgery research, there are still numerous unanswered questions regarding the associations between premorbid psychological factors and postoperative headache (POH) after microsurgical treatment via retro sigmoid approach for vestibular schwannoma (VS).
the General comments:
The spelling, grammar, and punctuation are very good. There are no errors that need to be corrected.
Abstract
The abstract is concise. All the necessary information about the study is included. The reader is fully prepared to understand the manuscript.
Introduction
- The information provided in the introduction is important for the comprehension of the article.
- The objective of the study is clearly mentioned in the last paragraph.
Methods
- The study design is well explained and is of good quality.
Results
- The results are presented in a very extensive way.
- The table is really helpful and necessary for the completion of the author's work.
Discussion
- The discussion is of great quality and includes updated data.
- I would suggest that the authors inform thoroughly the reader about the study's limitations (such as retrospective study, small sample)
Conclusion
From the presented data, the conclusion is complete and represents the work that the authors did.
References
The reference list covers the relevant literature adequately and in an unbiased manner
Author Response
Dear Reviewer,
We appreciate the time and effort that you and the other reviewers have dedicated to providing your valuable feedback on my manuscript.
I restructured the results section due to comments from another reviewer and added some more information to the discussion as you suggested. All the changes I made are highlighted in the manuscript.
We look forward to hearing from you in due time regarding our submission and to respond to any further questions and comments you may have.
Round 2
Reviewer 1 Report
Accept in present form